# The Associations between Racial Disparities, Health Insurance, and the Use of Amputation as Treatment for Malignant Primary Bone Neoplasms in the US: A Retrospective Analysis from 1998 to 2016

**DOI:** 10.3390/ijerph19106289

**Published:** 2022-05-22

**Authors:** Hans Lapica, Matan Ozery, Harsha Raju, Grettel Castro, Pura Rodriguez de la Vega, Noël C. Barengo

**Affiliations:** 1Department of Translational Medicine, Herbert Wertheim College of Medicine, Florida International University, 11200 SW 8th Street, Miami, FL 33199, USA; hlapi001@fiu.edu (H.L.); mozer003@med.fiu.edu (M.O.); hraju001@med.fiu.edu (H.R.); gcastro@fiu.edu (G.C.); rodrigup@fiu.edu (P.R.d.l.V.); 2Department of Health Policy & Management, Robert Stempel College of Public Health & Social Work, Florida International University, 11200 SW 8th Street, Miami, FL 33199, USA

**Keywords:** primary bone neoplasm, osteosarcoma, amputation, limb salvage, race, racial disparity

## Abstract

Primary bone neoplasms (PBNs) represent less than 1% of diagnosed cancers each year. Significant treatment disparities exist between racial and ethnic groups. We investigated patients with PBNs to determine an association between race/ethnicity and procedure-type selection. A non-concurrent cohort study was conducted using the SEER database. Patients diagnosed with PBNs between 1998 and 2016 were included (*n* = 5091). Patients were classified into three racial groups (Black, White and Asian Pacific Islanders) and were assessed by procedure-type received. The outcome was amputation. Race was not associated with increased amputation incidence. Hispanic patients had a 40% increased likelihood of amputation (OR 1.4; 95% CI 1.2–1.6). Insurance status was an independent predictor of procedure selection. Uninsured patients were 70% more likely to receive amputation than insured patients (OR 1.7; 95% CI 1.1–2.8). We recommend provider awareness of patients less likely to seek regular healthcare in the context of PBNs.

## 1. Introduction

Primary bone neoplasms (PBNs) represent less than 1% of total diagnosed cancers each year; however, due to non-specific symptoms, delayed presentation, and lack of suspicion from physicians, they are associated with significant morbidity and mortality [1]. The majority of these cases present in childhood with a small increase in incidence occurring later in life in people aged 60 years and older [1,2]. The majority of PBNs are non-metastatic at initial presentation, but can eventually metastasize, primarily to the lungs and other bones, highlighting the importance of early recognition and diagnosis [1,2,3]. Before 1970, surgical amputation was the only treatment for osteosarcoma and, even with amputation, mortality due to metastatic disease was still high at 80% [1,3]. With technological advances over the past several decades and the development of neoadjuvant chemotherapies and limb-salvaging surgical resection, the number of amputations has substantially decreased, and most primary bone neoplasms today are treated with limb-salvaging resection and reconstruction [1,2,3,4,5].

Amputation is a functionally disabling and life-changing treatment associated with physical and mental consequences. Some studies have investigated the differences in outcomes between amputation and limb-salvaging surgery and found no difference in post-operative local recurrence between patients who received amputation versus limb-salvage surgery [5,6,7,8]. Other studies have shown that patients receiving amputation treatment have significantly lower 5-year overall survival rates [1,5,7,8,9] and increased rates of metastasis [5,8] when compared with patients receiving limb-salvage surgery. The amputation rates for the treatment of various ailments have been shown to differ between different races and ethnicities, and the association between Black race and increased likelihood of amputation as a treatment for a variety of illnesses is well documented. Several causal factors for these disparities have been isolated, including increased disease severity on admission, less overall access to healthcare, and poorer physician to physician communication [10].

Despite this established association between Black race and increased amputation rates, there are few studies that analyze this association specifically in the treatment of malignant primary bone neoplasms. PBNs are a group of diseases where, for a long time, severely debilitating amputation was the standard of care. Considering the rarity of these neoplasms, it is also important to consider that these diseases are not well studied in general. Existing studies discuss the association of amputation with Black race in the context of peripheral vascular disease and diabetes. Additional studies have been conducted on sarcomas in general due to the lack of patients with primary bone neoplasms, and many discuss the disparate mortality rates associated with cancer treatment. Overall, many of the studies found had a smaller sample size in comparison to our study.

The objective of our study was to assess the incidence of amputation and limb-salvage surgery in the treatment of primary bone neoplasms across different races and ethnicities. This study therefore elucidates the disparate rates of amputation over more desirable resection with limb preservation and provides valuable information to clinicians on rare conditions that are not frequently studied but can be encountered in practice. 

## 2. Materials and Methods

This study utilized a non-concurrent cohort design and consisted of a secondary analysis of data obtained from the 2014 SEER (Surveillance, Epidemiology, and End Results Program) research data records.

The SEER database covers approximately 28% of the US population and includes data on populations with various specified cancers. Our study population was composed of US subjects with primary bone neoplasms who underwent amputation or limb-salvage surgery between 1975 and 2014, with associated data taken from the SEER database. We included patients 15 years or older with primary bone neoplasms (ICD-O-3 SEER Histological Type 8000/3, 8800/3, 8810/3, 9180/3, 9220/3, 9250/3, 9260/3) who had received surgical treatment, with the requirement that both race and surgical type was specified within the SEER database (Surgical Site Code 40–41.9). Patients were excluded if they did not meet these requirements. 

The main independent variable assessed in our study was racial status categorized into Black, White, and Asian/Pacific Islander. Our outcome variables included two major categories of surgical treatment for primary bone neoplasms. The outcomes, based on ICD codes for surgery types (surgtype_R1), were categorized as either major amputation/amputation or local tumor destruction/excision, including radical excision/resection with limb salvage. Surgery types included under major amputation/amputation included partial amputation of the limb, total amputation of the limb, forequarter amputation including the scapula, hindquarter amputation including the ilium/hip bone, and hemipelvectomy. The surgery types classified under the limb-salvage category included local tumor destruction or excision, partial resection/internal hemipelvectomy, and radical excision or resection of the lesion with limb-salvage.

The socio-economic variables that were included in the analysis were insurance status and poverty status. Demographic information such as age and sex were also included. Age was categorized in increments of 15 years and sex was categorized as male or female. Ethnicity was dichotomized into non-Hispanic and Hispanic (which included Mexican; Puerto Rican; Cuban; South or Central American excluding Brazil; other specified Spanish/Hispanic origin; Spanish/Hispanic/Latino not otherwise specified; NHIA surname match only; and the Dominican Republic). Geographic regions were divided by state including New York, Massachusetts, Connecticut, Georgia, Kentucky, Wisconsin, Iowa, Louisiana, New Mexico, Utah, Idaho, Washington, California, Alaska, and Hawaii. Grade of the tumor was divided into grades I, II, III, and IV. Grade I is well-differentiated and differentiated, grade II refers to moderately differentiated, grade III refers to poorly differentiated, and grade IV refers to undifferentiated or anaplastic. 

Stata was used to analyze the data. Frequency distributions of categorical variables descriptive of our population were assessed while controlling for data missing from SEER due to inconsistent reporting. Measures of central tendency such as mean were used in the descriptive analysis. Bivariate analysis (chi-square tests) was used to assess the presence of confounders. Collinearity diagnostics were used to assess the correlation between the variables. Both adjusted and unadjusted logistic regression analysis was used to calculate the odds ratios (OR) and 95% confidence intervals (CI). Covariates adjusted for in the multivariate regression analysis included: sex; age; ethnicity; tumor size; tumor grade; geographical region; insurance status; poverty level; decade of diagnosis; and surgical primary site. *p* values of <0.05 were considered statistically significant.

## 3. Results

Table 1 presents the baseline characteristics of the patient population by race. Most of the patients were White (*n* = 4125). The rest of the patients were either Black (*n* = 581) or of other races (*n* = 385), which consisted of Asian, American Indian, and Pacific Islander races. White patients had the largest percentage identified as Hispanic (25.5%, *p* < 0.001). They also had the largest proportion of patients in the 65–102-years-old age group (11.2%, *p* < 0.001). When compared with Black patients and others, they had the smallest proportion of patients with poorly differentiated (17.2%, *p* < 0.001) and undifferentiated neoplasms (29.9%, *p* < 0.001). Compared with Whites and others, Black patients had a higher percentage that were uninsured (4.8%) or enrolled in Medicaid (37.2%, *p* < 0.001). Black patients also comprised the largest proportion of undifferentiated malignancies (40.6%, *p* < 0.001). The other race category had the highest proportion of patients who were insured (74.8%, *p* < 0.001), followed closely by Whites (72.4%, *p* < 0.001). The other covariates did not differ statistically significantly according to race.

Table 2 presents the unadjusted and adjusted associations between participant characteristics and odds of amputation. The unadjusted models revealed that racial and ethnic disparity exists between race and amputation. Races other than Black had increased odds of amputation compared with White race (OR 1.2; 95% CI 1.0–1.5). Hispanic ethnicity had 1.4-fold higher odds of amputation than non-Hispanic patients (95% CI 1.2–1.6). The first statistical model was adjusted for all covariates except insurance status, as that variable was not collected before 2007. The second adjusted model was run including insurance for the years 2007 and onwards to check if insurance status confounded the association between race and amputation. After adjustment, race was not statistically significantly associated with odds of undergoing amputation as a treatment of choice. However, being of Hispanic ethnicity was statistically significantly associated with 40% increased odds of amputation (95% CI 1.2–1.6) when compared with non-Hispanics. After adjusting for insurance status, the association between race and amputation rates remained statistically insignificant. Hispanics maintained a 1.4-fold increase in odds of undergoing amputation (95% CI 1.1–1.7) when compared with their non-Hispanic counterparts. Of note, patients that were uninsured (OR 1.7; 95% CI 1.1–2.8) or were enrolled in Medicaid (OR 1.4, 95% CI 1.2–1.8) had significantly higher odds of undergoing amputation when compared with insured patients. Additionally, patients in the 65–102-years-old age group showed a 1.6-fold increased likelihood of undergoing amputation (OR 1.6; 95% CI 1.1–2.2) when compared to patients in the 26–49 age group. When compared to patients with well-differentiated tumors, patients with moderately differentiated tumors were associated with a 1.5-fold increased likelihood of amputation (95% CI 1.1–2.0), while poorly differentiated tumors were associated with a 2.5-fold increased likelihood of amputation (95% CI 1.8–3.3), and undifferentiated tumors were associated with 2.4-fold increased odds of amputation (95% CI 1.8–3.1). After adjusting for insurance, moderately differentiated tumors showed a 10% reduction in odds of amputation (OR 1.4; 95% CI 0.9–2.0), while poorly differentiated (OR 2.0, 95% CI 1.3–3.0) and undifferentiated tumors (OR 1.9; 95% CI 1.3–2.8) both showed a 50% reduction. Finally, a more recent year of diagnosis was associated with lower odds of amputation when compared with a diagnosis between 1998 and 2001. A diagnosis from 2002 to 2005 or 2006 to 2009 showed a 30% reduction (95% CI 0.60–0.84), while a diagnosis from 2010 to 2013 was associated with a 40% reduction (95% CI 0.50–0.74) and a diagnosis from 2014 to 2016 showed a 50% reduction in odds of amputation (95% CI 0.40–0.63).

## 4. Discussion

Our data revealed no statistically significant association between White, Black or other races and amputation. However, Hispanic ethnicity increased the odds of receiving amputation as a treatment for primary bone neoplasms over limb-salvage surgery compared with non-Hispanics. Our data also found that male sex, age greater than 65, regional and distant cancer stage, advanced cancer grades, and lack of insurance or Medicaid enrollment were independent predictors of amputation, while a more recent year of diagnosis decreased the likelihood.

With respect to race, this result is generally inconsistent with the findings of the current scientific literature. Black race has been shown to be associated with increased rates of amputation for the treatment of various limb-compromising diseases such as diabetes mellitus (DM), peripheral arterial disease (PAD), and soft-tissue sarcomas (STS) [10,11,12]. Two studies specifically looked at amputation and limb-salvage rates for patients with osteosarcoma. Downing et al. found that patients of Black race were less likely to undergo limb-salvage procedures than non-Black patients [13]. Moreover, the majority consensus has demonstrated increased mortality based on race even in studies with no difference in rate of procedure type [12,13,14,15,16]. However, Traven et al. reported no difference in amputation vs. limb-sparing procedure rates based on race alone [15]. This agrees with another study that did not find a difference in procedure-type based on race; however, this was looking at soft-tissue sarcomas only in insured patients [16]. There are several possible explanations for these inconsistent findings. In general, the studies investigating STS and PBN were more likely to find no difference in amputation rates based on race than the studies investigating amputation rates in diseases with direct vascular compromise.

We hypothesize that due to the slowly progressive nature of malignant neoplasms compared to the acute vascular effects of DM and PAD, the consequences of delayed access to healthcare may be partially reduced. Additionally, in our study, the largest proportion of Hispanic patients was within White race, our largest racial group, while Black race had a substantially smaller sample size.

We did find a significant difference in amputation rates based on ethnicity, where Hispanics were more likely to undergo amputation than non-Hispanics. This is consistent with previous studies demonstrating a decreased likelihood of limb-salvage surgery in Hispanic patients [10,17,18,19]. In two of these studies, the investigators looked at both race and ethnicity and found that patients of Black race and Hispanic ethnicity were more likely to be amputated for the complications of PAD and lower leg infections [10,20], while Martinez et al. found that only Hispanic patients had increased rates of limb salvage for the treatment of STS [17].

Previous studies have proposed that the higher rates of amputation in minority groups may be due to greater participation in adverse health risk behaviors, decreased compliance to treatment, weakened trust in their healthcare providers, and more advanced states of disease due to less access to healthcare [11,12]. Insurance status is an important metric for quantifying an aspect of socioeconomic status and access to healthcare. We hypothesize that this lack of access to quality health insurance plays a direct role in the increasing odds of amputation in these minority groups by causing a delay in cancer treatment which allows the primary bone neoplasms to progress to more advanced stages and grades that are directly associated with higher rates of amputation.

When a primary bone neoplasm is allowed to grow without intervention, it may grow to a size which is incapable of being excised with large-enough margins or it may grow into vital vascular supply, necessitating amputation for the survival of the patient [1,2,3]. Although the patient may live due to amputation, they ultimately will have lower 5-year survival, decreased functional scores, and higher metastatic recurrence [5,7,8,9]. This results in decreased life expectancy and quality of life in contrast to a patient who is treated earlier and, therefore, is more likely to receive limb-salvage surgery [21]. Osteosarcoma and Ewing sarcoma are the two most common primary bone neoplasms included in our study. These neoplasms tended to have bimodal age distributions reflected in our “0–25” and “>50” age groups; however, we found the likelihood of amputation to be significantly higher in the older age groups. We attribute this difference to age-related differences in healing capacity and the integrity of vascular supply required for successful limb-salvage surgery [1].

Our study found that there was a statistically significant association between both lack of insurance and Medicaid enrollment and increased likelihood of amputation. There are few studies investigating race and treatment outcomes for primary bone neoplasms [13,16,17], but only one that included insurance status in the analysis by studying a population of solely insured patients [16]. Pak et al. looked at patients within the TRICARE insurance database and did not include patients 65 or older due to Medicare. When controlling for the variable of insurance status, they reported that there was no difference in amputation rates between White and Black race for the treatment of osteosarcoma [16]. Although our study included insured patients, uninsured patients, patients enrolled in Medicaid, and patients older than 65, we similarly found no difference in amputation rates based on race, even with adjustments for insurance status and age.

Naturally, our study has some limitations. The main limitation of our study is missing data not recorded in the SEER database [19]. For instance, insurance status was unavailable until 2007, reducing the sample size of the study by 45%. However, after adjusting for insurance, the association between amputation and both race and ethnicity did not change. Other missing information that may have influenced amputation rates and could not be adjusted for included: any pre-existing comorbidities, histological response to chemotherapy, genetics, and familial cancer syndromes in patients, as well as the specific location of tumors or involvement of any major vessels, nerves, or joints. Furthermore, we did not have access to information about the study participants’ socioeconomic status, which could act as an additional barrier to healthcare and further affect the need for amputation. To remedy this, we used health insurance status as a proxy for socioeconomic status. Additionally, SEER only collects data from certain geographic locations and states, excluding participants from most of the US. This limits the data collected and the generalizability to the rest of the US population of patients with primary bone neoplasms. Finally, many forms of primary bone neoplasms are treated in different ways depending on the location, size, stage, and type of the tumor. Furthermore, the incidence of different types of bone neoplasm varies according to race and ethnicity. Unfortunately, we were unable to control for these factors as that information was not available in the SEER database.

## 5. Conclusions

In conclusion, the knowledge obtained from this study reinforces many previous hypotheses on disparities in treatment for diseases where limb loss is a common consequence. We recommend that physicians be more cognizant and aware of the prevalence of these “rare” diseases, particularly with patients who do not regularly seek healthcare, because the cost of missing a diagnosis and delaying treatment is too severe.

Although quality of life measurements are decreased for patients undergoing any type of invasive surgical procedure, amputation is associated with even worse outcomes [5,7,8,9]. Newer surgical techniques and neo-adjuvant therapies are decreasing the prevalence of amputation every year [4], yet we still find consistently higher relative rates in minority populations. We support a call to action to the legislators of this country to expand and strengthen access to quality insurance to address the disproportionate presence of minority groups who are either uninsured or on Medicaid.

This issue goes far beyond the scope of primary bone neoplasms. These proposed explanations for increased morbidity and mortality in minority groups mirror the explanations suggested by investigators studying the lethal complications for more common diseases such as peripheral arterial disease and diabetes mellitus. The etiologies that lead to limb loss are countless but, for the majority, the solution is identical. Considering that physicians often overlook these rare diagnoses, it is important to ensure that other issues such as race, ethnicity, and socioeconomic factors do not compound the risk of a missed diagnosis. Future research should investigate the complex relationship between race, ethnicity, and insurance status, and further highlight the additional socioeconomic factors that can influence healthcare outcomes in the setting of potentially severe rare diseases. With rare diseases specifically, it is essential that all possible case data are extracted and maintained properly so that future secondary data analyses have sufficient statistical power to make confident evidence-based conclusions to improve existing guidelines on treatment.

## Figures and Tables

**Table 1 ijerph-19-06289-t001:** Characteristics of study participants with primary bone neoplasms who received surgical treatment according to race from 1998 to 2016.

	White	Black	Other	*p*-Value
	(*n* = 4125)	(*n* = 581)	(*n* = 385)	
	%	%	%	
**Ethnicity**				<0.001
Non-Hispanic	74.5	97.1	96.9	
Hispanic	25.5	2.9	3.1	
**Age (years)**				<0.001
0–25	51.3	59.9	60.8	
26–49	23.2	24.3	22.1	
50–64	14.2	9.5	10.4	
65–102	11.2	6.4	6.8	
**Sex**				0.226
Male	57.2	53.5	56.7	
Female	42.8	46.5	43.3	
**Stage**				0.684
Localized	39.8	39.8	39.6	
Regional	46.4	44.6	46.3	
Distant	13.9	15.6	14.1	
**Insurance**				<0.001
Uninsured	3.4	4.8	2.1	
Medicaid	24.2	37.2	23.1	
Insured	72.4	57.9	74.8	
**Grade**				<0.001
Well differentiated	12.7	8.9	8.6	
Mod. differentiated	16.8	14.5	12.5	
Poorly differentiated	17.2	19.6	20.3	
Undifferentiated	29.9	40.6	36.4	
Unknown	23.4	16.4	22.3	
**Rural**				0.033
Non-rural	98.6	99.3	100	
Rural	1.4	0.7	0	
**Year of Diagnosis**				0.171
1998–2001	14.7	13.4	13.5	
2002–2005	21.8	23.4	19.2	
2006–2009	22.8	22.6	22.1	
2010–2013	22.8	22.7	20.8	
2014–2016	17.9	17.9	24.4	

**Table 2 ijerph-19-06289-t002:** Unadjusted and adjusted associations between participant characteristics and odds of amputation.

Characteristics	Unadjusted	Adjusted	Adjusted Insurance
	OR ^1^ (95% CI ^2^)	OR (95% CI)	OR (95% CI)
**Race**			
White	Ref. ^3^	Ref.	Ref.
Black	1.1 (0.9–1.3)	1.1 (0.9–1.4)	1.1 (0.8–1.5)
Other	1.2 (1.0–1.5)	1.3 (1.0–1.7)	1.3 (1.0–1.9)
**Ethnicity**			
Non-Hispanic	Ref.	Ref.	Ref.
Hispanic	1.4 (1.2–1.6)	1.4 (1.2–1.6)	1.4 (1.1–1.7)
**Age**			
0–25	1.3 (1.1–1.5)	0.9 (0.8–1.1)	0.9 (0.7–1.2)
26–49	Ref.	Ref.	Ref.
50–64	1.2 (1.0–1.5)	1.2 (0.9–1.5)	1.3 (0.9–1.8)
65–102	1.4 (1.1–1.8)	1.4(1.1–1.8)	1.6 (1.1–2.2)
**Sex**			
Male	1.4 (1.2–1.6)	1.3(1.1–1.5)	1.3 (1.0–1.5)
Female	Ref.	Ref.	Ref.
**Stage**			
Localized	Ref.	Ref.	Ref.
Regional	1.8 (1.6–2.1)	1.6 (1.3–1.8)	1.6 (1.3–2.0)
Distant	2.4 (2.0–3.0)	2.0 (1.6–2.5)	2.0 (1.5–2.7)
**Insurance**			
Uninsured	1.6 (1.0–2.5)	N/A	1.7 (1.1–2.8)
Medicaid	1.5 (1.3–1.9)	N/A	1.4 (1.2–1.8)
Insured	Ref.	Ref.	Ref.
**Grade**			
Well differentiated	Ref.	Ref.	Ref.
Mod. differentiated	1.6 (1.2–2.1)	1.5 (1.1–2.0)	1.4 (0.9–2.0)
Poorly differentiated	2.6 (2.0–3.4)	2.5 (1.8–3.3)	2.0 (1.3–3.0)
Undifferentiated	2.7 (2.1–3.4)	2.4 (1.8–3.1)	1.9 (1.3–2.8)
Unknown	1.8 (1.4–2.4)	1.7 (1.3–2.3)	1.5 (1.0–2.3)
**Year of Diagnosis**			
1998–2001	Ref.	Ref.	Ref.
2002–2005	0.7 (0.60–0.84)	0.7 (0.5–0.8)	N/A
2006–2009	0.7 (0.60–0.85)	0.7 (0.5–0.8)	N/A
2010–2013	0.6 (0.50–0.74)	0.6 (0.5–0.7)	0.9 (0.7–1.1)
2014–2016	0.5 (0.40–0.63)	0.4 (0.3–0.5)	0.7 (0.5–0.8)

^1^ Odds ratio (OR); ^2^ Confidence interval (CI); ^3^ Reference (Ref).

## Data Availability

National Cancer Institute: Surveillance, Epidemiology, and End Results (SEER) Available: http://seer.cancer.gov, accessed on 13 November 2016.

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
