# Peer review of "The Associations between Racial Disparities, Health Insurance, and the Use of Amputation as Treatment for Malignant Primary Bone Neoplasms in the US: A Retrospective Analysis from 1998 to 2016"

_ijerph, 2022, doi:10.3390/ijerph19106289_

Round 1

Reviewer 1 Report

In this manuscript, the authors investigated the incidence of amputation and limb-salvage surgery in the treatment of primary bone neoplasms across different races and ethnicity. I would like to put forward the following questions and comments:

  1. In Table 2, it shows unadjusted and adjusted associations between participant characteristics and odds of amputation. Why both unadjusted and adjusted associations were used? Please give more explanation on them.  
  2. Beyond the limitation mentioned in the manuscript, there are many kinds of primary bone neoplasms, and bone neoplasms are treated in different ways depends on the location, size, stage and type. And the incidence of different types bone neoplasm varies in different races and ethnicity. These factors were not taken into consideration.
  3. In the manuscripts, the authors discussed a lot about the relation between insurance status and amputation, which digresses from its title.

Author Response

Thank you very much for all your valuable comments and guidance to improve our manuscript. We have carefully studied your comments and revised our work according to your suggestions. Your feedback helps us to become better researchers and are highly appreciated.

Comment#1: In Table 2, it shows unadjusted and adjusted associations between participant characteristics and odds of amputation. Why both unadjusted and adjusted associations were used? Please give more explanation on them. 

Response#1:  We agree with the reviewer that we need to provide more information on the interpretation of the unadjusted value for race and amputation. The unadjusted logistic regression analysis reveals whether a racial disparity exist, thus, the unadjusted models are important. The unadjusted model also helps to identify possible confounders of the association between race and amputation. Once a racial disparity is identified in the unadjusted analysis, the adjusted analysis reveals whether this disparity can be explained by the covariates included in the model or not. Therefore, we believe it is important to show both models. To clarify this, we have added the following sentences to the result section:

“The unadjusted models revealed that racial and ethnic disparity exist between race and amputation. Race other than black had increased odds of amputations compared with white (OR 1.2; 95% CI 1.0-1.5). Hispanic ethnicity had 1.4-fold higher odds of amputation than non-Hispanic patients (95% CI 1.2-1.6).”

Comment#2: Beyond the limitation mentioned in the manuscript, there are many kinds of primary bone neoplasms, and bone neoplasms are treated in different ways depends on the location, size, stage, and type. And the incidence of different types of bone neoplasm varies in different races and ethnicity. These factors were not taken into consideration.

Response#2: We agree with the reviewer that this is an important limitation. We have added the following sentences to the limitation section to address it:

Finally, the many forms of primary bone neoplasms are treated in different ways depending on the location, size, stage, and type of the tumor. Furthermore, the incidence of different types of bone neoplasm varies according to race and ethnicity. Unfortunately, we were unable to control for these factors as that information was not available in the SEER database.

Comment#3: In the manuscripts, the authors discussed a lot about the relation between insurance status and amputation, which digresses from its title.

Response#3:  Thank you for this excellent observation. We have changed the title of the manuscript to provide a better idea what is discussed in the manuscript. The title is now as follows:

The Associations between Racial Disparities, Health Insurance, and the Use of Amputation as Treatment for Malignant Primary Bone Neoplasms in the US: A Retrospective Analysis from 1998-2016.

Reviewer 2 Report

dear authors,
the proposed article sought to assess the existence of a relationship between race / ethnicity and type of treatment between different racial and ethnic groups. I find the article interesting and the results provided quite valid, however some corrections are needed:
- In the results section, the process of correcting the data should be better clarified
- Table 2 reports data regarding the year of diagnosis, however this is not mentioned in the text
- In the discussion on line 167 the authors state: one year of more recent diagnosis has reduced the probability .... explain why, in the results no mention is made of this thing
- The discussion should be summarized because it could be long and boring for the reader
- in the discussion, in line 215 the authors state: osteosarcoma and Edwing's sarcoma were the two most common primary bone neoplasms included in the study .... in the results no mention is made of this. it would be appropriate to describe the types of bone neoplasms included in the results section.

-in the discussion on line 169 add references

Author Response

Thank you very much for all your valuable comments and guidance to improve our manuscript. We have carefully studied your comments and revised our work according to your suggestions.

Comment#1: In the results section, the process of correcting the data should be better clarified

Response#1: We have now provided information in regard the variable health insurance and why we did not include it in the first model.

Following sentence was added to the result section to clarify how the model was corrected for health insurance:

The first statistical model was adjusted for all covariates except insurance status as that variable was not collected before 2007. The second adjusted model was run including insurance for the years 2007 and onwards to check if insurance status confounded the association between race and amputation.

Comment#2: Table 2 reports data regarding the year of diagnosis, however this is not mentioned in the text

Response#2: We agree with the reviewer and have added the following sentence to the description of the results:

Finally, a more recent year of diagnosis was associated with lower odds of amputation when compared with a diagnosis between 1998 and 2001. A diagnosis from 2002-2005 and 2006-2009 showed a 30% reduction (95% CI 0.60-0.84), while a diagnosis from 2010-2013 was associated with a 40% reduction (95% CI 0.50-0.74), and a diagnosis from 2014-2016 showed a 50% reduction in odds of amputation (95% CI 0.40-0.63).

Comment#3: In the discussion on line 167 the authors state: one year of more recent diagnosis has reduced the probability .... explain why, in the results no mention is made of this thing

Response#3: As mentioned above, we have now added the following sentence to the description of the results in order to be consistent with the main summary of the findings of our study (first paragraph in the discussion section.

Finally, a more recent year of diagnosis was associated with lower odds of amputation when compared with a diagnosis between 1998 and 2001. A diagnosis from 2002-2005 and 2006-2009 showed a 30% reduction (95% CI 0.60-0.84), while a diagnosis from 2010-2013 was associated with a 40% reduction (95% CI 0.50-0.74), and a diagnosis from 2014-2016 showed a 50% reduction in odds of amputation (95% CI 0.40-0.63).

Comment#4: The discussion should be summarized because it could be long and boring for the reader

Response#4: We agree with the reviewer and have shortened the discussion and the conclusion section. The changes are marked in the text. We hope this improved the readability.

Comment#5:  In the discussion, in line 215 the authors state: osteosarcoma and Edwing's sarcoma were the two most common primary bone neoplasms included in the study .... in the results no mention is made of this. it would be appropriate to describe the types of bone neoplasms included in the results section.

Response#5: In order follow your advice to shorten the discussion, we have removed that part from the discussion section. Moreover, this was not a main finding/objective of our study.

Comment#6: In the discussion on line 169 add references

Response#6: That part is a summary of the main results of our study, thus, does not refer to previous studies. Therefore, no reference is provided. For clarity, we have now pointed out clearly that this is the summary of our findings. The paragraph states now as follow:

Our data revealed no statistically significant association between White, Black, and other race and amputation. However, Hispanic ethnicity increased the odds of receiving amputation as treatment for primary bone neoplasms over limb-salvage surgery, compared with non-Hispanics. Our data also found that male sex, age greater than 65, regional and distant cancer stage, advanced cancer grades, and lack of insurance or Medicaid enrollment were independent predictors of amputation, while a more recent year of diagnosis decreased the likelihood.

Round 2

Reviewer 1 Report

The authors have give reasonable explanation to my questions and revised the manuscript accordingly.